

# Tsunami Vulnerability Mapping of Coastal Areas to Confront the Banda Detachment Tsunami (Case Study at Tual City)

Viona Dika Rikumahu[1]

1School of Architecture, Planning and Policy Development, Bandung Institute of Technology, Bandung, 40132, Indonesia

*Correspondence to*: Viona Dika Rikumahu (Vionadika56@gmail.com)

**Abstract.**

Tsunami is a natural disaster that is most feared by humans, especially for people who live in coastal areas that are directly adjacent to seismically active seas in Indonesia. Dullah Island in Tual City is located in a coastal area directly adjacent to the Banda Sea, making this area have a high level of tsunami hazard. The Banda Sea is known as an active sea that emits earthquake
energy, both those that do not have the potential to cause a tsunami. The existence of previous research on the run-up height of tsunami waves in Tual City makes it important to study the characteristics of the Tual City area to formulate disaster mitigation-based planning using the GIS (Geographic Information System) based qualitative spatial analysis method with variables of slope, height of the area, distance from the coastline, and land use. The high level of tsunami hazard and extremely high population exposure considering that the majority of the population in the coastal area lives on the coast, it is important
to study the characteristics of the region to formulate disaster mitigation-based planning to create a *Build Back Better* region and community

## 1 introduction

Tsunami is a very destructive and potentially hazardous natural phenomenon, and can trigger multidimensional disasters such as building damage, environmental damage, health crises, famine, and social chaos (Retno Susilorini et al., 2021). Therefore,
adequate mitigation and preparation are essential in the face of tsunami disasters, including through education and socialization to the community on how to recognize and avoid the hazards of tsunamis, as well as the development of disaster-resistant infrastructure (Horspool et al., 2014).

The Banda Sea is part of the collision zone between the edge of the Australian continent and the Banda arch. This arc is a deformation line of overlapping folds and faults, which is why several earthquakes in 1674 and 1708 triggered tsunamis in
Ambon, Haruku, Buru and surrounding areas (Harris & Major, 2017).

Coastal areas are dynamic places where biological, chemical and geological features change rapidly. Due to their dynamics, coastal areas have high natural resource potential, but are also prone to natural hazards derived from land and sea, such as floods and tidal waves, tsunamis, and storms. Therefore, coastal areas are vulnerable to natural disasters, but also have immense potential for a variety of economic and livelihood activities (Robin Davidson-Arnott).





The existence of V-shaped beaches or bays that narrow inland means that tsunamis can easily travel deeper inland, thereby increasing the potential damage and danger to people living in these areas. For that reason, a good knowledge of coastal morphology and tsunami risk mitigation is critical to protect the population in coastal areas with these morphological characteristics (Heru Sri Naryanto).

Based on the journal tsunami modelling around the Banda sea and the implications of affected inundation,the largest inundation
result is 1009.49 meters at the research location (Rahmawati et al., 2017), furthermore, the physical characteristics of the research area are not explained. The objective of this article is to show how the physical conditions of the affected area using GIS (Geographic Information System) analysis with research variables in the form of land elevation maps, slopes, distance from the coastline, settlements then overlaid using GIS. To Based on the journal tsunami modelling around the Banda sea and the implications of affected inundation,the largest inundation result is 1009.49 meters at the research location (Rahmawati et
al., 2017), furthermore, the physical characteristics of the research area are not explainedimprove the effectiveness of risk management and disaster mitigation today, the use of remote sensing satellites and Geographic Information Systems (GIS) has become a well-integrated and successful tool in disaster research (Sambah et al., 2018). The structure of this article consists of the background, the analysis method used, the results of the research in the form of a tsunami exposure level map.

## 2 Data and Methods

### 2.1 Study Area

The research area is Dullah island, Tual City. Dullah island, Tual City is located in Southeast Maluku regency which is geographically located at the coordinates 131° - 132° East longitude and 5° 32' -8° 00' South latitude. Administratively, Tual City is located southeast of the capital of Maluku province (Ambon), with borders to the north with Banda sea, south with Arafura sea, west with Banda sea and east with Nerong strait (Southeast Maluku Regency). With a population of 93,145 people
and an area of 235.38 km². Surrounded by the ocean, especially in the north and west, namely the Banda Sea, the city of Tual is often shaken by earthquakes on both large and small scales.



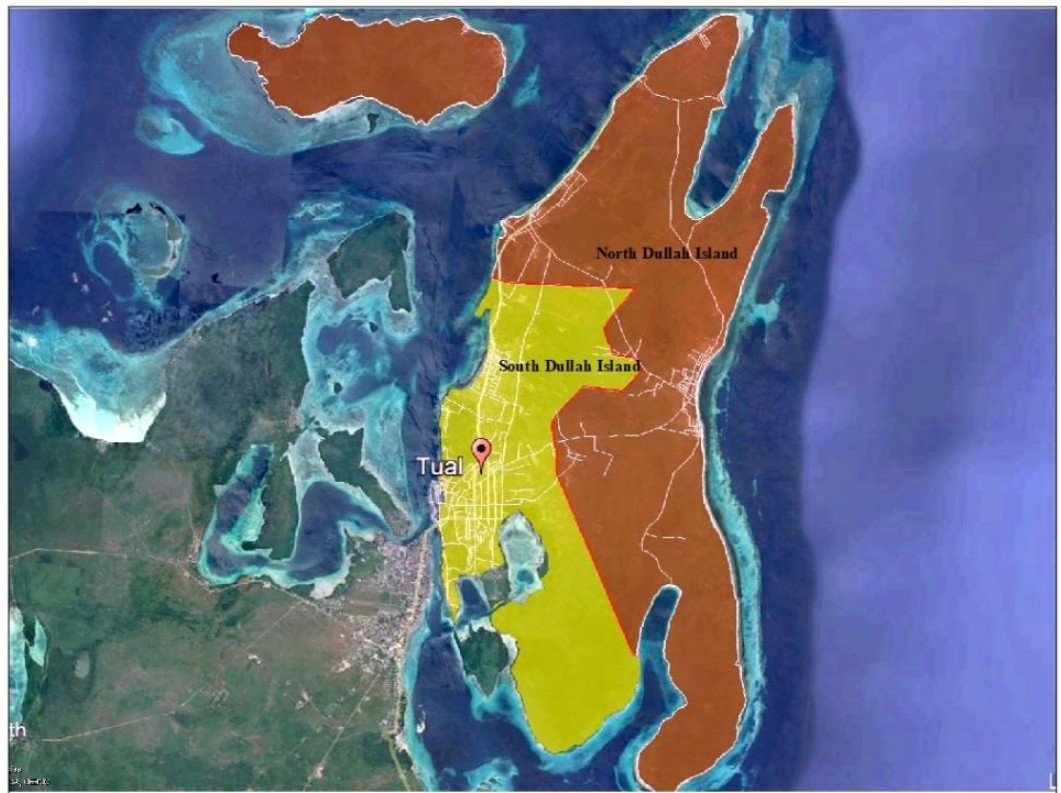

**Figure 1: Map of the Study Area, Showing Dullah Island, Tual City with Administrative Boundaries.**

© Google Earth

2.2 Research Flows

The data collection process in this research uses secondary data from Dem-National, RBI map (Indonesia disaster risk), google earth and Tual City in numbers. The data used in this research consists of data on slope, altitude, coastline, land use, river line. In making the tsunami hazard map there are 4 variables used, namely data on distance from the coastline, elevation map, distance from the river and land use. Data on distance from the coastline was accessed through www.tanahair.indonesia.go.id.

The elevation and slope map of the research area was obtained from the management of ASTER (The Advanced Spaceborne Thermal Emission and Reflection Radiometer) GDEM (Global Digital Elevation Model) Image accessed through the web page from Https://www.earthexplorer.usgs.gov/. After analysing the tsunami vulnerability variables through Arcgis, the tsunami hazard map was overlaid with the settlement map to see how the settlements in Tual City were exposed to tsunami hazards. The resulting data was then processed, analysed, and interpreted using descriptive spatial analysis. To show how the tsunami

exposed areas in Tual City are.



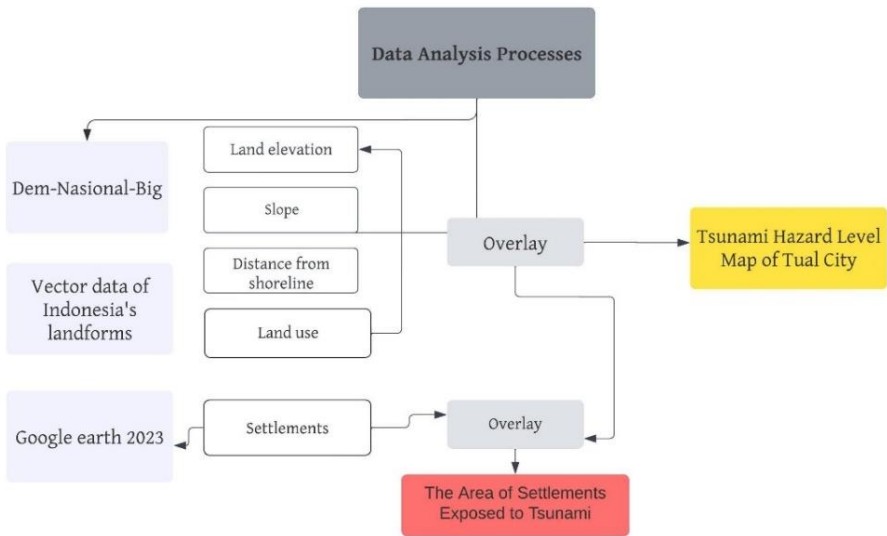

Figure 2: Stages of Data Analysis

To determine the tsunami hazard level of an area and its settlements, parameters are needed to be further analysed through

arcgis, these parameters are generated through a literature study adjusted to the existing conditions of the research area.

| Parameters | Value | Classification | Score | Percentage |
|---|---|---|---|---|
| Slope % (Iqoh,2013) | 0-8 | Very high | 5 | |
| | 8-15 | high | 4 | |
| | 15-25 | moderate | 3 | 25 |
| | 25-45 | low | 2 | |
| | >45 | Very low | 1 | |
| Land elevation (Lida 1963) | 0-12 | Very high | 5 | |
| | 13-30 | high | 4 | |
| | 31-45 | moderate | 3 | 25 |
| | 46-64 | low | 2 | |
| | >64 | Very low | 1 | |
| Distance from shoreline | 0-500 | Very high | 5 | |
| | 501-1000 | high | 4 | |
| | 1001-1500 | moderate | 3 | 30 |
| | 1501-3000 | low | 2 | |
| | >3000 | Very low | 1 | |
| Land use | Settlement, Forest, Marsh, River, Rice Fields | Very high | 5 | |
| | Agriculture / Land Vegetation | high | 4 | 20 |
| | Field/moor | moderate | 3 | |
| | Bushes, Lakes, Reeds | low | 2 | |
| | Forest, Limestone, Rock | Very low | 1 | |

Table 1. Parameters to Measure Tsunami Hazard Level



## 3. Results

Beach shape: the shape of the bay will experience a multiple amplification/increase in wave energy. A headland shape will

experience a reduction in wave energy. Single island and will experience impact from the side"

(Diposaptono & Budiman, 2008). The character of Dullah island of Tual City can be described as follows:

a) Topography of Dullah Island, Tual City which is flat or contour with a slope of 8%.

b) The shape of the beach: hugging and strait will experience an increase in wave energy, Dullah Island, Tual City in the north and south are surrounded by the Bay Lav.

| Parameter | Classification | Area | |
|---|---|---|---|
| | | Km² | Percentages |
| **Slope** | Very gentle slope | 52,68 | 49,84 |
| | Gentle slope | 37,54 | 35,52 |
| | Moderately sleep slope | 13,34 | 12,62 |
| | Steep slope | 1,98 | 1,87 |
| | Escarpments | 0,15 | 0,14 |
| **Elevation** | 0-12 | 12,08 | 11,34 |
| | 13-30 | 60,33 | 56,65 |
| | 31-45 | 23,73 | 22,28 |
| | 46-64 | 9,35 | 8,78 |
| | >64 | 1 | 0,94 |
| **Distance from shore line** | 0-500 | 34,36 | 16,09 |
| | 501-1000 | 24,71 | 11,57 |
| | 1001-1500 | 15,61 | 7,31 |
| | 15001-3000 | 6,51 | 3,05 |
| | >3000 | 132,38 | 61,98 |
| **Land use** | Lake | 0,59 | 0,26 |
| | Dense forest | 109,98 | 48,98 |
| | Agriculture | 71,62 | 31,89 |
| | settlement | 6,95 | 3,09 |
| | Bushes | 30,06 | 13,39 |
| | Bare soil | 0,55 | 0,24 |
| | Moor | 4,81 | 2,14 |

Table 2. Results of Parameter Analysis to Identify The Level of Tsunami Vulnerability





a. Slope

From the results of the map analysis, it can be seen that the island of Dullah, Tual City, is dominated by a very gentle area with a slope vulnerability of 0 - 8%, the area is in a very high vulnerability hazard zone. While the slope of all areas on Dullah Island is included in the very high tsunami hazard zone this is because the area has a very gentle slope There is a close correlation between slope vulnerability and the level of tsunami hazard. If the area has a steep slope, the tsunami wave height will be lower and the resulting hazard will be lower. Otherwise, if the area has a shallow slope, the height of the tsunami waves The slope vulnerability parameter has a significant impact on the extent of the tsunami that will flood the land.

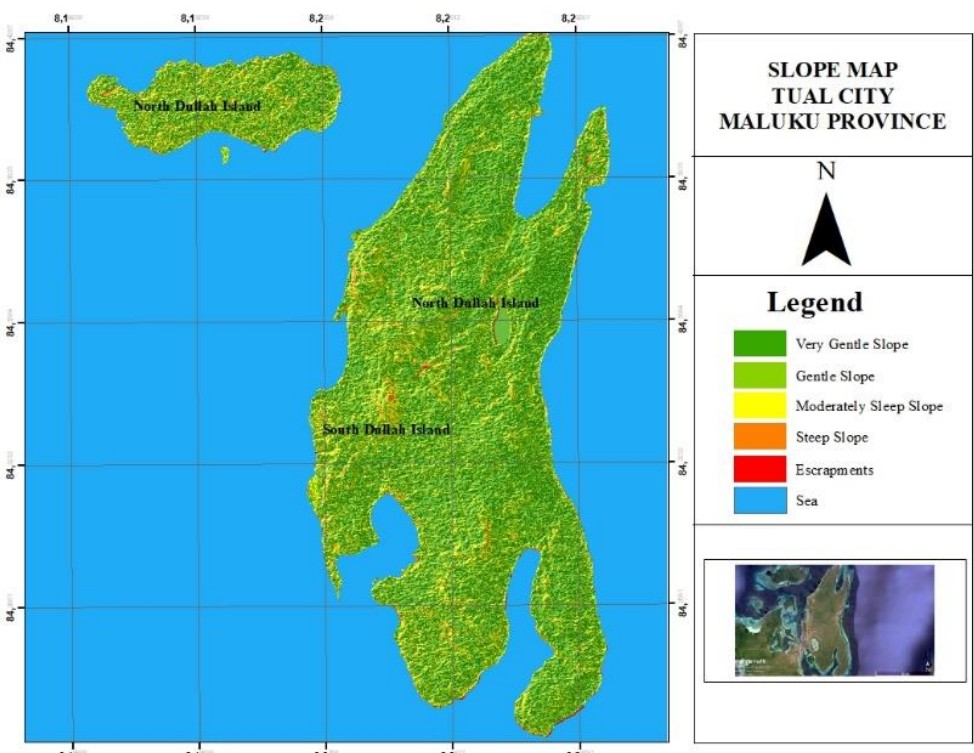

Figure 3: Slope Map

© Analysis Results By The Author

b. Land elevation

The mapping of land elevation vulnerability levels in this study is divided into five vulnerability categories as can be seen in the map below. Based on the analysis conducted, Dullah Island, Tual City is dominated by an elevation of 13-30 metres, which means that the area has a high vulnerability to tsunami disaster. While the area with an elevation of 31 - 45 metres is an area with a moderate level of vulnerability, this area is also still included in the danger zone based on the height of the surface in the Dullah Island area, Tual City.



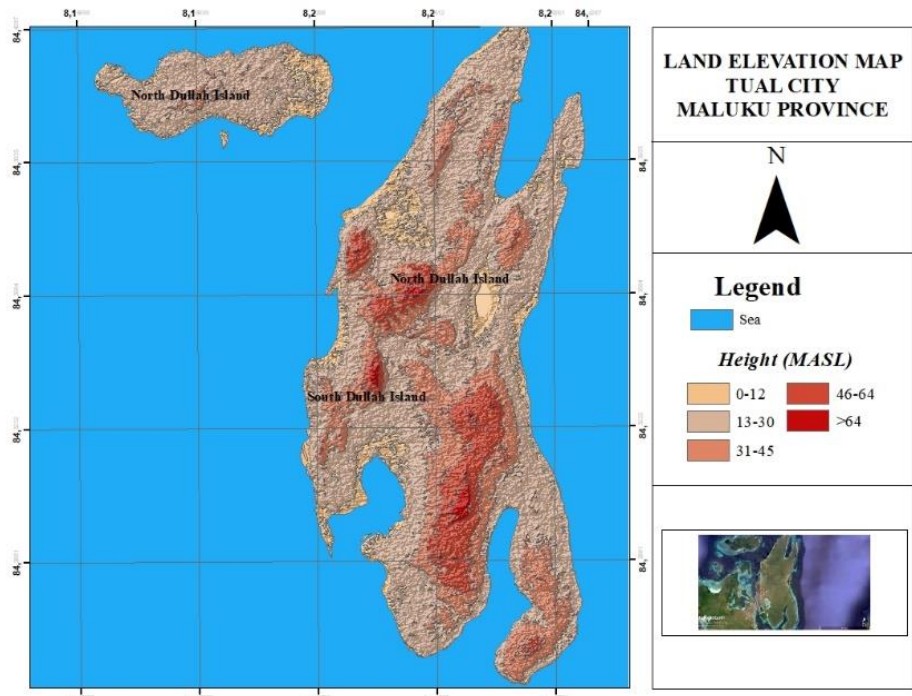

100                                 Figure 4: Land Elevation

© Analysis Results By The Author

c. Distance From Shoreline

The distance from the coastline indicates the level of vulnerability of an area to tsunami hazards, in other words, the closer a place is to the coastline, the greater the risk of tsunami hazards because tsunami propagation is a complex
105        physical phenomenon (Sabri, 2014). As the distance from the coastline increases, the wave reach and height decrease. There are five classes for measuring tsunami hazard using the variable distance from the coastline. Areas with the highest level of hazard are located between 0-500 metres and >3000 metres from the coastline. Based on the map above, Dullah island of Tual City is located above 3000 metres above sea level with a presentation of 61.98% with an area of 132.38 km.



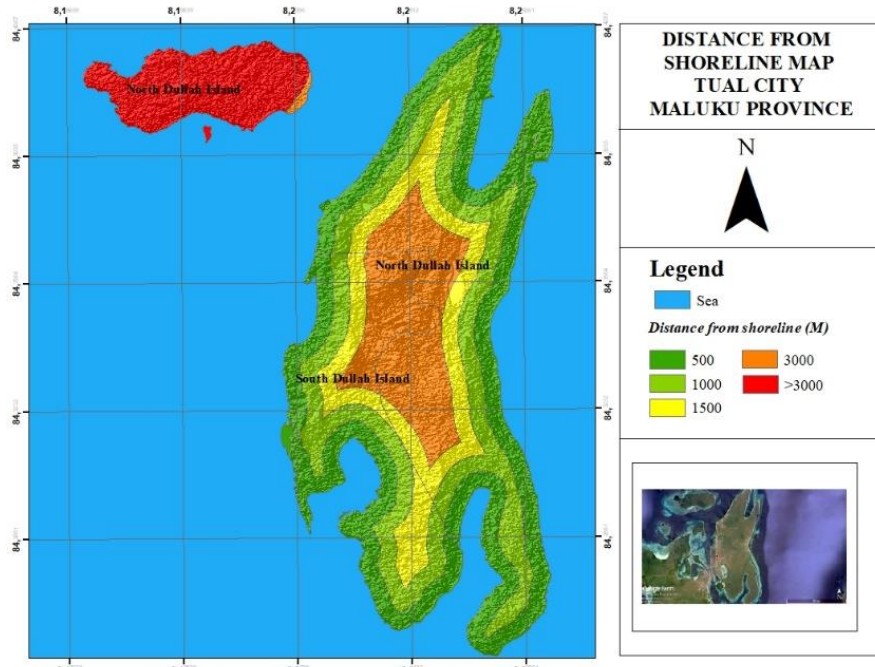

Figure 5: Distance From Shoreline

© Analysis Results By The Author

d. Land use

The classification of the land use parameters is as follows: Residential land is categorised as ``highly threatened'', Agricultural land is categorised as ``vulnerable'', and Undeveloped land is categorised as ``highly threatened''.Forest is categorised as ``not endangered''. The hazard class is based on the level of risk and loss due to tsunami disaster. According to Prawiradisastra (2011), residential and urban areas are land use types that have a high risk of tsunami disaster. These areas are considered highly vulnerable due to losses caused by damage to buildings and other settlements. Land use in the Tual City area is dominated by agriculture, which falls into the prone category.



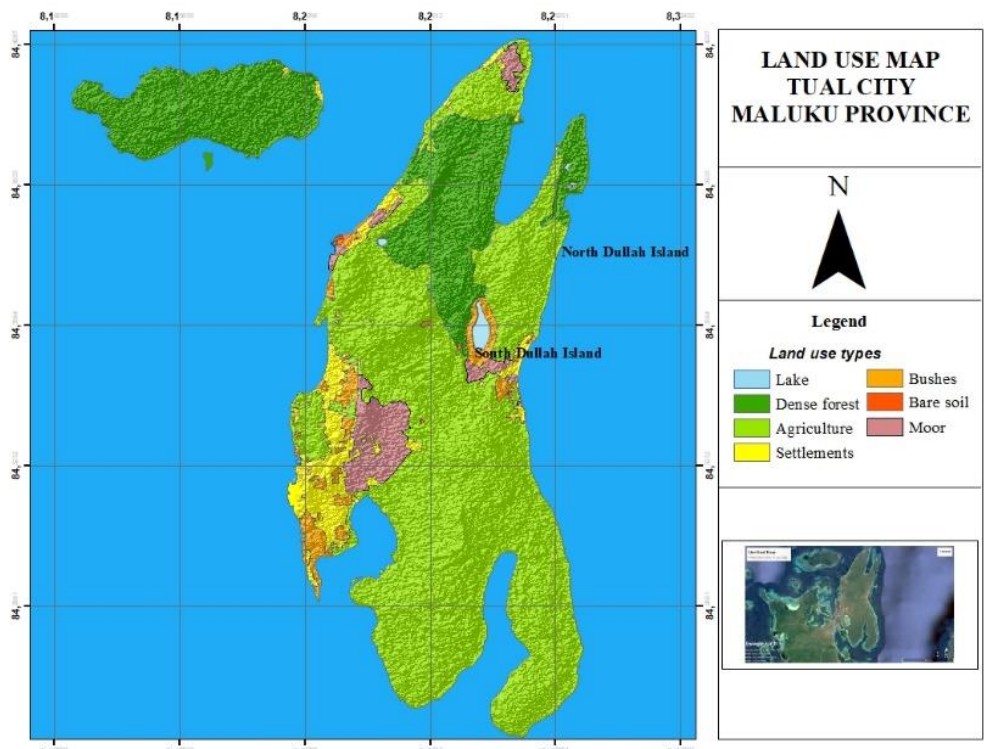

Figure 6: Land Use Map

© Analysis Results By The Author

e.  Tsunami Vulnerability Map

| Hazard class | Area (km²) | Tsunami Vulnerability Percentages |
|---|---|---|
| High | 75,79 | 71,78 |
| Moderate | 29,35 | 27,80 |
| Low | 0.45 | 0.42 |

Table 3. Tsunami Vulnerability Map Show Hazard Classification, Area, and Percentages

The level of tsunami disaster risk in tsunami prone areas on Dullah Island, Tual City, is classified as high to moderate. The high tsunami disaster risk level is located in the sub-districts of Dullah Utara and Dullah Selatan, as 130    well as the moderate tsunami disaster risk level. So the distribution of high and moderate tsunami risk is almost equally distributed in both sub-districts. With a high parameter percentage of 71.78% and a medium of 27.80%.



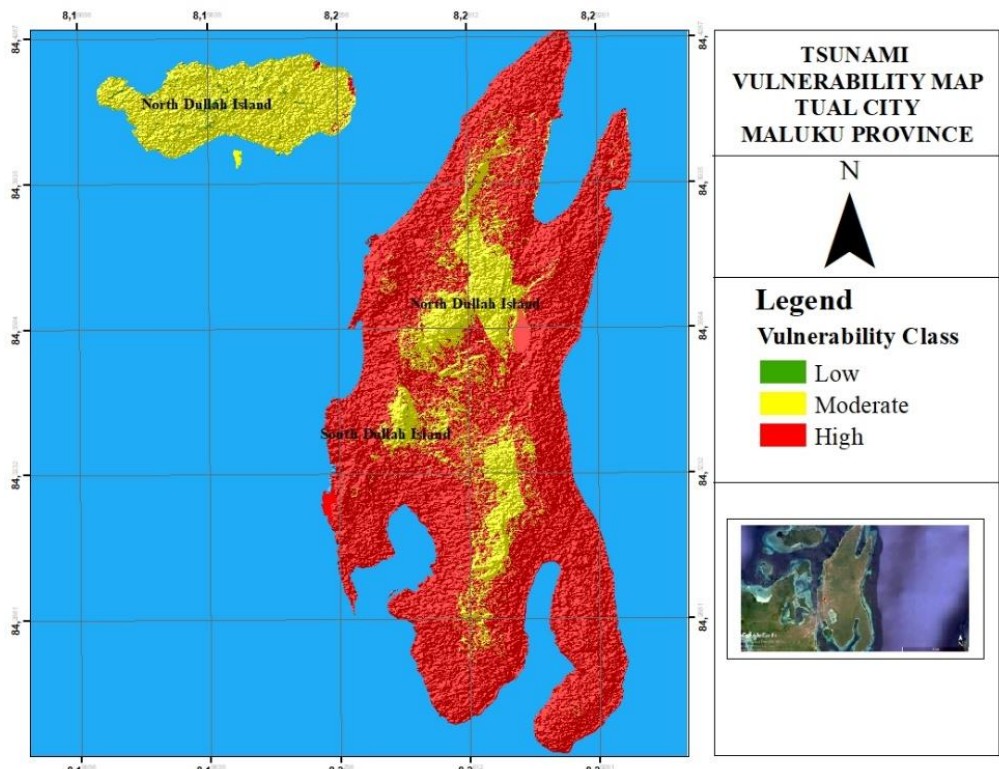

Figure 7: Tsunami Vulnerability Map

© Analysis Results By The Author


## 4. Conclusion

Tsunami Exposure of Settlements

| Hazard class | Area(Hectare) | Tsunami hazard percentage of settlement |
|---|---|---|
| High | 897,58 | 94,38 |
| Moderate | 53,58 | 5,60 |
| Low | 0,2 | 0,02 |

Table 4. Tsunami Exposure of Settlements

Based on the tsunami hazard map of Tual City, it can be concluded that Tual City has a high level of tsunami vulnerability. And the area affected by 94.38 per cent is almost all settlements in Pullau Dullah. If considering the existing conditions of the area, most of the settlements in Pullau Dullah are located on the coast and Pullau Dullah itself borders the sea so that the presentation of the affected settlements is also getting bigger.

The high level of tsunami vulnerability is also influenced by the slope, the height of the area, and the shape of the city of 145 Tual like a crab claw where the location of the bay on the north and south sides of the area is able to accommodate the



energy of the tsunami waves released. The location of the city of Tual is surrounded by oceans, especially the western and northern parts which are surrounded by the Banda Sea, where the Banda Sea is one of the most sismically active seas in Indonesia.

looking again at research done by (Pownall et al., 2016), from Australia's National University in his journal entitled Rolling
Open Earth's Deepest Forearc Basin, he found that a 7 km deep ravine beneath the Banda Sea in eastern Indonesia, has been formed by an extension structure mechanism, along what is probably the largest identified fault plane on Earth. The Banda and Weber Deep detachments are among the largest of their kind on modern Earth, comparable in scale to "fossilised" examples preserved in older terranes. These faults have caused many large earthquakes as well as tsunami threats for areas immediately adjacent to the Banda Sea, such as Tual City.

It is important for mitigation-based planning using the build back better method for areas in eastern Indonesia, especially Maluku, which borders the Banda Sea, to be conscious of the high vulnerability of the area to tsunamis, both from understanding the physical conditions of the area as studied in this journal and other conditions such as social, economic and ecological conditions, in order to develop a tsunami disaster mitigation strategy.

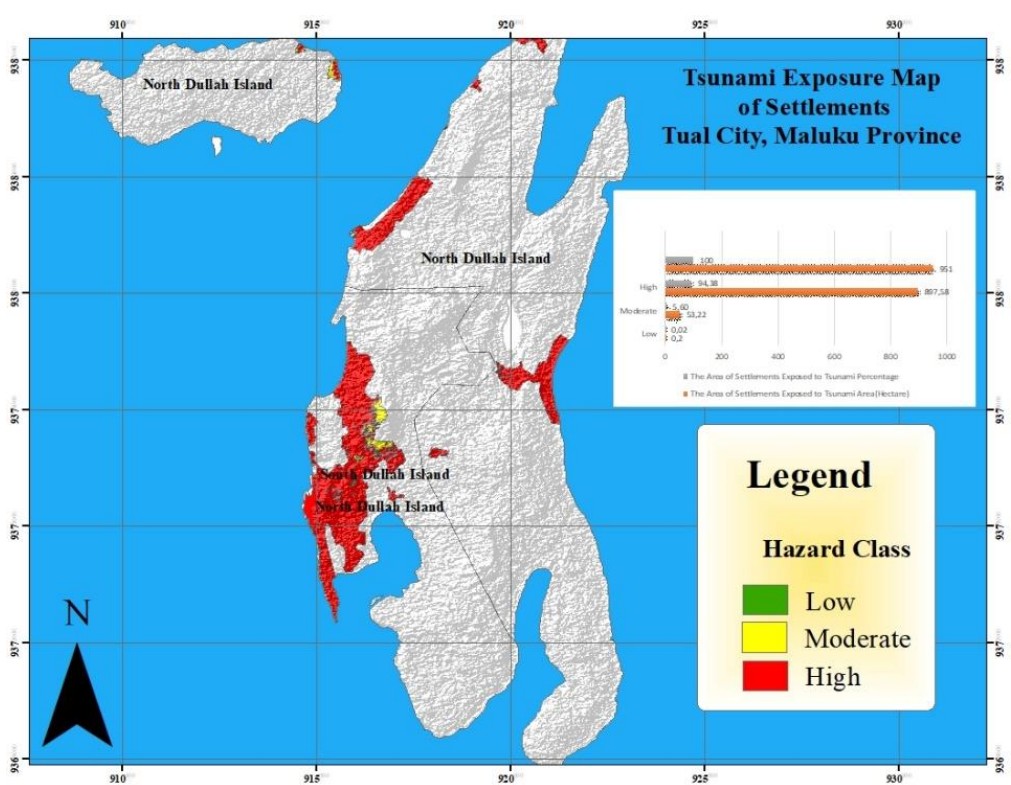

160            Figure 8: Tsunami Exposure Map of Settlements

            © Analysis Results By The Author

**5. Competing Interests**

The contact author has declared that none of the authors has any competing interests.




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
