# Peer review of "Tsunami Vulnerability Mapping of Coastal Areas to Confront the Banda Detachment Tsunami (Case Study at Tual City)"

_EGUsphere, 2023_

## Referee Comment (RC1)

**Review**

On the manuscript **"Tsunami Vulnerability Mapping of Coastal Areas to Confront the Banda Detachment Tsunami (Case Study at Tual City)"** by Viona Dika Rikumahu.

**Overview: Conditional Acceptance Upon Minor Revision**

The paper presents data analysis of the mapping of several physical variables (through GIS) believed by the author to affect the wave run-up of tsunami. The method is then applied to a case study of a small island in Indonesia that is subject to significant seismicity due to the location between the Banda Arch and the Australian plate. The results are sound and provide an interesting case study and contribute to advancing the knowledge of local properties of natural hazards. However, below you will find minor specific comments (including the references therein) that should be addressed/added towards the publication of this work:

1. The title is too long and confusing. I suggest the author rewrites it in the form of **"Tsunami Vulnerability Mapping of Coastal Areas: A case study of/from the Banda Sea"**

2a. In line 8 the author writes *"Dullah Island in Tual City..."* but the Island can not be within the city, rather the opposite, the city is located in the Dullah Island.

2b. In lines 9-10 the author writes *"Dullah Island in Tual City is located in a coastal area directly adjacent to the Banda Sea, making this area have a high level of tsunami hazard. The Banda Sea is known as an active sea that emits earthquake energy, both those that do not have the potential to cause a tsunami."* First, the second sentence should be written first, and only then you can claim *"making this area have a high level of tsunami hazard".* Secondly, there are incorrect terms in the second sentence, such as *"emits earthquake energy".* Thirdly, it is followed by a contradiction, with *"both those that do not have the potential to cause a tsunami"* which I believe both was meant to be but. Be more careful with the grammar and vocabulary. I suggest therefore that you rewrite this part as **"The Banda sea is an active seismic region. Since Tual City, being located in Dullah Island, has coastal areas directly adjanced to the Banda sea, it makes this area..."**.

2c. Although it is obvious to the expert, it would be good to clarify to which slope you refer to in line 13, which I believe to be the **beach slope**.

2d. Again contradiction between *"The high level of tsunami hazard"* in line 13 with *"both those that do not have the potential to cause a tsunami"* in line 10. I Belive the writing in line 10 is a mistake.

2e. Remove the word *"potentially"* in line 18.

3a. Lines 20-22 provide an insufficient introduction to the physical phenomenon of tsunami. For instance, the author failed to describe basic features, such as that tsunami is a water wave created by disturbances in the water column, which several types exist. The author could cite recent cases such as Sumatra 2004, Tohoku 2011, and most recently Tonga 2022. Cite briefly their characteristics from theoretical and phenomenological papers.

3b. [Suggestion for 3a] The introduction could be better formulated and widened for a broad spectrum of readers by discussing how coastal and geomorphological processes affect tsunami run-up heights. The author should also reference a few articles that describe the relevance of each variable used in table 1 in the introduction. Disturbance on tsunami and water wave signals by local bathymetry and coastal topography, resonance, and other processes should be cited (Pelinovsky and Mazova, 1992; Pelinovsky and Poplavsky, 1996; Kânoğlu and Synolakis, 1998; Pelinovsky et al., 2001; Pelinovsky, 2006; Golinko et al., 2006; Tsai et al., 2013; Rybkin et al., 2014; Park et al., 2015; Shimozono, 2016; Chugunov et al., 2020; Li et al., 2021; Mendes

et al., 2022; Mendes and Kasparian, 2022; Li and Chabchoub, 2023).

3c. Line 23 you write *"Australian continent"* but I believe you refer to the **"Australian plate"** instead.

3d. Lines 26-29 in the first page should go up to the start of the introduction, together with my suggestions in comments 3a and 3b.

3e. Line 30 you discuss v-shaped bays, please add references and discuss physical and theoretical reasoning for their relevance to tsunami. These lines should also go up at the start of the introduction together with 3a,3b,3d.

3f. In line 34 the author writes *"journal tsunami modelling"*. Is this a typo? To me it looks like you tried to write **"Based on tsunami modelling around the Banda sea"**. Furthermore, you should be more specific and write which type of modelling (analytical, empirical, numerical), what type of solver (in case of numerical), and cite references.

3g. Between lines 35-36 you write *", furthermore, the physical characteristics of the research area are not explained."* This is improper english and quite confusing. From my understanding of this phrase, you should have written **"However, the physical characteristics of beach shape and inland morphology were not considered in the latter study"** referring to Rahmawati et al. 2017.

3h. In line 36 you should start the second sentence with **"Consequently, the objective..."**

3i. Lines 38-40 seem to be a copy of lines 34-35, delete them. The new sentence seems to start at the wrong justaposition *"explainedimprove"*, which means the sentence starts with *"improve"*. Of course this is a small mistake unspotted by the author which can easily be fixed. Please rephrase it.

3j. In line 51 the author talks about large and small scales, please add references.

4a. The title of subsection 2.2 is confusing. The term *"Research Flows"* seem to be a weird alternative to **"Methodology"**. Please change this to a widely used term.

4b. Line 59 discusses the effects of four variables. Their effect should be carefully delineated, it is up to the author to decide whether the full description should appear here or in the introduction. However, both the introduction and section 2.2 should have an explanation for why you chose these four variables.

4c. Lines 59 and 62 have https links in the text. This is a bad practice. The author should either add footnotes and place the links in them, or write them as references and put the links in the reference section.

4d. In line 63 the author uses the term *"overlaid"* when referring to a map. This sentence does not explain which figure it is overlaid. It is possible that the author talks of the method in general, which then I suggest to change the tense of the sentence, for instance: **" After analysing the tsunami vulnerability..., a tsunami hazard map is/ will be overlaid with the..."**

4e. In table 1 we see what seems to be two references (Iqoh, 2013 and Lida, 1963) that are not to be found in the reference section. Fix that.

4f. You should not write slope units in %, because the slope can exceed 100%.

4g. Write down the units for land elevation and distance from the shoreline for both table 1 and table 2.

5a. Line 74 - This is not how you start a section. Write a short sentence saying that after implementing the methodology there are several outcomes. Firstly, the beach shape...

5b. Although you had in table 1 classification for the slopes and ranges thereof, in table 2 you changed the names of the classifications. Either make the same name in tables 1/2, or add slope ranges for table 2.

5c. In the classification for slopes of table 2 there is the typo **"sleep"** within *"Moderately sleep slope"*→**"Moderately steep slope"**.

**Conclusion**

The reviewer thanks for the opportunity to read this important work. Overall, I support the publication of this preprint once all these minor issues have been clarified/amended.

**References**

Chugunov, V.A., Fomin, S.A., Noland, W., Sagdiev, B.R., 2020. Tsunami runup on a sloping beach. Computational and Mathematical Methods 2, e1081.

Golinko, V., Osipenko, N., Pelinovsky, E., Zahibo, N., 2006. Tsunami wave runup on coasts of narrow bays. International Journal of Fluid Mechanics Research 33.

Kânoğlu, U., Synolakis, C.E., 1998. Long wave runup on piecewise linear topographies. Journal of Fluid Mechanics 374, 1–28.

Li, Y., Chabchoub, A., 2023. On the formation of coastal extreme waves in water of variable depth. Cambridge Prisms: Coastal Futures , 1–11.

Li, Y., Draycott, S., Zheng, Y., Lin, Z., Adcock, T., Van Den Bremer, T., 2021. Why rogue waves occur atop abrupt depth transitions. Journal of Fluid Mechanics 919, R5.

Mendes, S., Kasparian, J., 2022. Saturation of rogue wave amplification over steep shoals. Phys. Rev. E 106, 065101.

Mendes, S., Scotti, A., Brunetti, M., Kasparian, J., 2022. Non-homogeneous model of rogue wave probability evolution over a shoal. J. Fluid Mech. 939, A25.

Park, H., Cox, D.T., Petroff, C.M., 2015. An empirical solution for tsunami run-up on compound slopes. Natural Hazards 76, 1727–1743.

Pelinovsky, E., 2006. Hydrodynamics of tsunami waves, in: Waves in geophysical fluids: tsunamis, rogue waves, internal waves and internal tides, pp. 1–48.

Pelinovsky, E., Mazova, R.K., 1992. Exact analytical solutions of nonlinear problems of tsunami wave run-up on slopes with different profiles. Natural Hazards 6, 227–249.

Pelinovsky, E., Poplavsky, A., 1996. Simplified model of tsunami generation by submarine landslides. Physics and Chemistry of the Earth 21, 13–17.

Pelinovsky, E., Talipova, T., Kurkin, A., Kharif, C., 2001. Nonlinear mechanism of tsunami wave generation by atmospheric disturbances. Natural Hazards and Earth System Sciences 1, 243–250.

Rybkin, A., Pelinovsky, E., Didenkulova, I., 2014. Nonlinear wave run-up in bays of arbitrary cross-section: generalization of the carrier–greenspan approach. Journal of Fluid Mechanics 748, 416–432.

Shimozono, T., 2016. Long wave propagation and run-up in converging bays. Journal of Fluid Mechanics 798, 457–484.

Tsai, V.C., Ampuero, J.P., Kanamori, H., Stevenson, D.J., 2013. Estimating the effect of earth elasticity and variable water density on tsunami speeds. Geophysical Research Letters 40, 492–496.

---

## Referee Comment (RC2)

**NHESS manuscript egusphere-2023-3021-- Tsunami Vulnerability Mapping of Coastal Areas to Confront the Banda Detachment Tsunami (Case Study at Tual City)**

This study presents an approach to establishing the tsunami hazard map of Tual City, which is undoubtedly an interesting and meaningful research topic. However, despite its potential, I find myself unable to support the publication of this manuscript due to several key issues:

1.  The entire paper is poorly organized and written. The introduction, in particular, lacks comprehensiveness and persuasiveness.

2.  The title is not appropriate.

3.  The abstract requires further improvement. It should provide background information, address central questions/research gaps, outline research goals, summarize main findings/results, and emphasize the significance of the findings.

4.  Duplicated sentences have been identified between lines 34 to 45, indicating a need for careful revision and editing.

5.  Numerous grammar errors and typos are present throughout the manuscript.

6.  In lines 58 and 59, the use of four variables to create the hazard map is mentioned. However, the rationale behind selecting these specific variables is unclear. I recommend that the authors provide clarification or cite relevant references to justify their choice.

7.  The flowchart of data analysis depicted in Figure 2 is confusing, making it difficult to understand how the data is processed and analyzed across different blocks. Clarification is needed.

8.  The values in columns 4 and 5 in Table 2 do not seem to be correct. Additionally, the unit for area should be "km²" for consistency.

9.  In Line 86 on Page 6, the authors claimed that "There is a close correlation between slope vulnerability and the level of tsunami hazard". However, this claim lacks support from associated results.

10. Despite the inclusion of figures and tables in the manuscript, there is a lack of clear references to them within the context of the discussion.